# Validating self-reported exclusive breastfeeding in Eswatini using stable isotope techniques

Kwanele Siyabonga Simelane[1]*, Tholakele Mhlanga[2], Nhlakanipho Sandziso Mkhaliphi[2], Siniketiwe Zwane[2], Glorious Dlamini[2], Henry Gadaga[1], Helen Mulol[3,4], Anna Coutsoudis[5]

1 Department of Environmental Health-Food Science Unit, Faculty of Health Sciences, University of Eswatini, Mbabane, Eswatini, 2 Eswatini National Nutrition Council, Ministry of Health, Mbabane, Eswatini, 3 Department of Paediatrics, University of Pretoria, Pretoria, South Africa, 4 Research Centre for Maternal, Fetal, Newborn and Child Healthcare Strategies, University of Pretoria, Pretoria, South Africa, 5 Department of Paediatrics and Child Health, University of KwaZulu-Natal, Durban, South Africa

* kwasiyas@gmail.com

## Abstract

Exclusive breastfeeding (EBF) prevalence at the population level is typically assessed through maternal or caregiver reports, which are prone to recall and social desirability biases. The deuterium oxide dose-to-mother (DTM) technique offers an objective alternative by administering a small amount of deuterium-labeled water to lactating mothers. This study compared the prevalence of EBF from maternal recall versus the DTM method in Eswatini and evaluated the strengths and limitations of both approaches in assessing infant feeding practices linked to malnutrition. A total of 102 mother-infant pairs were recruited from three regional hospitals and three health centers in Eswatini. EBF was assessed by maternal recall using the WHO 24-hour infant feeding recall method and objectively measured using the DTM stable isotope technique at three infant age intervals: 6 weeks to 2.4 months (n = 26), 2.5 to 4.0 months (n = 43), and 4.1 to 5.5 months (n = 33). Two cutoff values defined EBF status via the stable isotope method. Agreement between maternal recall and DTM measurements was analyzed using Kappa statistics. Maternal recall consistently overestimated EBF across all time points compared to the DTM method. Using Model 1 cutoffs, prevalences of EBF by maternal recall vs. DTM were 100% vs. 50% (≤2.4 months), 79.5% vs. 18.2% (2.5–4 months), and 71.9% vs. 18.8% (4.1–5.5 months). For Model 2, prevalences were 100% vs. 65.4%, 79.5% vs. 50%, and 71.9% vs. 43.8%, respectively. The difference between the two models is the cutoff that was used to determine EBF using the DTM method. Kappa analysis showed minimal agreement between methods at each time point. The DTM technique provides a more accurate and objective measure of EBF than maternal recall and is the gold standard for smaller sample studies. Maternal recall substantially overestimated exclusive breastfeeding prevalence, while the deuterium oxide technique yielded

**Data availability statement:** The study data is a property of the Ministry of Health, Government of Eswatini, and are not publicly available. Interested readers may request access to the dataset by contacting the Secretariat of the EHHRRB. The general contact email address for data requests, through which readers can obtain access on an ongoing basis, is "es@ehhrrb.org.sz". This address is managed by the Eswatini Health and Human Research Review Board (EHHRRB), the national health research ethics committee under the Eswatini Ministry of Health. It is actively monitored by the Board's administrator. Please reference Protocol Number EHHRRB057/2022 in all correspondence.

**Funding:** The author(s) received no specific funding for this work.

**Competing interests:** The authors have declared that no competing interests exist.

more reliable, objective assessments of infant feeding practices, highlighting the need to interpret recall data with caution due to overreporting.

## Introduction

Exclusive breastfeeding (EBF) for the first 6 months of life is widely recognized as a cornerstone of optimal infant feeding, providing all required nutrients and fluids while protecting against gastrointestinal and respiratory infections and related mortality [1]. EBF also offers important protection against common infectious diseases such as diarrhoea and respiratory infections and may reduce the risk of overweight and obesity later in life [2]. In addition, longer and more EBF has been associated with improved neurocognitive outcomes, including higher verbal intelligence and better verbal and spatial skills [3] and confers health benefits for mothers, such as lower risks of breast and ovarian cancer [4]. The first two years of life represent a critical window for fostering healthy growth, establishing positive behaviours, and preventing acute malnutrition and irreversible stunting, ensuring that children reach their full developmental potential [5]. The World Health Organization (WHO) recommends that babies be exclusively breastfed (i.e., an infant receives only breast milk, with no other liquids or solids, not even water, for the first six months of life, except for oral rehydration solution, vitamin/mineral drops, or medicines) after which complementary feeding must be introduced [6]. Exclusive breastfeeding (EBF) is a major protective factor against stunting, which is a well-established marker for poor child development [7–12]. It is therefore imperative for countries to routinely collect and interpret infant feeding and growth indicators data to monitor malnutrition trends toward country specific and global targets. This data is valuable to inform the development of evidence-based policies and programs which address feeding practices in particular the protection, promotion and support of EBF. As of 2022 in Eswatini, the prevalence of EBF 0–6 months old, measured through maternal recall assessments, was reported at 77.28%, while stunting among children under five years remained significant at 25% [13,14]. This highlights the need for an objective measure to better validate EBF fidelity and address malnutrition more effectively.

The deuterium oxide 'dose-to mother' (DTM) stable isotope method is one of the techniques used to precisely and objectively determine exclusivity of breastfeeding, which can then be compared to EBF data obtained from maternal recall. The technique determines the average daily amount of breast milk consumed by the baby over a period of 14 days [15]. The use of the technique is well documented from studies conducted across various countries including Bangladesh, Botswana, Cameroon, South Africa [16–19] and others, to compare the prevalence of EBF based on maternal recall with those measured using the stable isotope technique. In all the studies, the prevalence of EBF reported by mothers were consistently higher than those measured directly through the objective DTM method [19]. This paper describes and reports the assessment and findings of EBF fidelity in Eswatini by comparing the DTM stable isotope method with maternal recall. The main objective is to assess the validity of EBF measurement methods and to provide evidence that could inform improvements in national monitoring and policy strategies.

Eswatini is a landlocked lower middle-income country in the Southern African Region with a population of nearly 1.2 million people, with almost 45% of the population below the age of 30 years. Life expectancy has increased to approximately 64 years, up from 58 years reported in 2021 [20]. Nationally, 54.8% of the population lives below the national poverty line, with poverty concentrated in rural areas where about 70% of rural residents are poor [21]. Approximately 75.6% of the population resides in rural areas [22]. According to the 2021–2022 Multiple Indicator Cluster Survey (MICS) data [22], the country faces significant public health challenges, including high HIV prevalence at 24%, inequalities marked by a gini-coefficient of 55% and significant child malnutrition, with 25% of children under five years reported to be stunted, indicating chronic malnutrition, and 2.5% of children under five years wasted, reflecting acute malnutrition.

## Materials and methods

### Study design

A cross-sectional study was conducted among breastfeeding mother–infant pairs attending postnatal and child welfare services at six health facilities in Eswatini. During routine postnatal care, when mothers typically receive health education, the study team explained the study's purpose and procedures. Mothers who expressed interest were invited to a separate session for a detailed briefing and to provide informed consent. All participating women provided written informed consent for their own participation and assent for the participation of their infants.

The study was conducted in purposively selected sites included three regional hospitals—Mankayane Government Hospital, Piggs Peak Government Hospital, and Good Shepherd Hospital—and three health centres: Matsanjeni Health Centre, Mkhuzweni Health Centre, and Dvokolwako Health Centre. These facilities are considered high volume facilities, serving on average per month about 488 breastfeeding mothers with infants aged 6 weeks to 5.5 months. At national level, the study was implemented in collaboration with the Eswatini National Nutrition Council and senior Ministry of Health officials, who contributed to selection of these study sites, refinement of data-collection procedures and tools, and agreement on how results would be reported back to program stakeholders. This participatory process was intended to enhance the relevance and uptake of the findings for breastfeeding and child-nutrition policies.

The study eligibility criteria included mothers aged 15–49 years who were clinically well (no acute/chronic illness affecting breastfeeding), and were currently breastfeeding at the time of recruitment. Infants were required to be aged between 6 weeks and 5.5 months, born at term, and with a birth weight greater than 2.3 kg. Exclusion criteria included mothers who reported not breastfeeding, multiple births, infants with any condition known to interfere with breastfeeding and infants with chronic illnesses such as congenital heart disease or cerebral palsy [19]. These exclusions were critical to give a realistic assessment of breast milk intake using the deuterium ($^2$H) oxide DTM technique in healthy breastfeeding mothers with healthy singleton infants. A total of 130 mother-infant pairs were approached and of these, 102 met the inclusion criteria and were enrolled in the study.

### Sample collection

In order to determine breastfeeding quantities and practices, the deuterium oxide DTM method was employed. The lactating mothers were given an accurately weighed 30g oral dose of deuterium oxide ($^2$H$_2$O). The $^2$H tracer is transferred from the mother to the infant through breast milk, allowing for the quantification of breast milk intake by the infant. Saliva samples (2mL each) were collected from the mother and infant prior to dosing and on days 1, 2, 3, 4, 13 and 14 from the 10th of August 2022 to the 30 August 2022. Trained research staff collected samples using dental absorbent swabs, SalivaBio Oral Swab (SOS) kit. For infants, the swab was gently moved around the oral cavity to ensure adequate saturation, while mothers were instructed to chew the swab to achieve saturation. Once saturated, each swab was placed inside a syringe

barrel and compressed to extract the saliva into pre-labelled cryovials. This procedure was repeated as necessary until the required volume of 2 mL each was obtained for mother and infant.

## Sample analysis

The quantity of tracer (deuterium oxide) transferred from mother to infant through breast milk intake (breastfeeding) was determined using an Agilent 4500T series Fourier Transform Infra-Red (FTIR) spectrometer [15]. This is based on the principle that molecules can absorb light in the mid-infrared region of the electromagnetic spectrum, which affects the vibration of chemical bonds. FTIR takes advantage of this principle to detect the presence of different hydrogen isotopes in saliva samples. Water exhibits three vibrational modes of the O-H bond [15], and the energy of these vibrations depends on the mass of the atoms involved. When hydrogen ($^1H$) is replaced by deuterium ($^2H$), there is a shift in the absorption region [15]. One of the advantages of FTIR is its high sensitivity: microgram (µg) quantities can be routinely measured using a good-quality FTIR spectrometer. The deuterium enrichment values on the specified sampling days were determined by analyzing the six post-dose saliva samples and the baseline (pre-dose) saliva samples for mother and infant, which were then entered into an Excel spreadsheet for the DTM method. The Solver function was then employed to model deuterium enrichment in the saliva samples over a 14-day period for each mother-infant pair. The Solver function minimizes the sum of squared differences between the FTIR-measured values and the values predicted by the model. The model estimates allow quantification of breast milk intake and non-milk oral intake (NMOI – intake of water from sources other than breast milk), and the latter can be used to determine whether the mother was exclusively breastfeeding her infant.

In this study, two cutoff models were used to classify the EBF status objectively. Model 1 represented the original and more stringent classification which defined EBF as NMOI ≤ 25g/day, predominant breastfeeding as NMOI of 25-220g/day, and partial breastfeeding as NMOI > 220g/day [16]. Model 2 was developed using the NMOI cutoff (86.6) derived from a previous study that analyzed deuterium turnover in mother–infant pairs observed to be exclusively breastfeeding in Indonesia [23]. This cutoff, corresponding to the 90th percentile of the posterior distribution of NMOI in that population, was used to classify exclusive breastfeeding status in our dataset. In this model, predominant breastfeeding was defined as NMOI between 86.7-220g/day, while partial breastfeeding remained as NMOI > 220g/day.

## Maternal recall

Maternal recall was assessed using a questionnaire on infant feeding practices. Through an interview, mothers reported infant feeding practices over the previous 24 hours, detailing breast milk intake alongside any liquids, semi-solids, or solids given. Breastfeeding indicators were obtained through interviews with mothers, using a questionnaire adapted from the WHO/UNICEF Indicators for Assessing Infant and Young Child Feeding, Part 2 [24] at the onset of sample collection. The tool is based on current guidelines regarding breastfeeding, complementary feeding, and the feeding of non-breastfed infants and young children under two years of age [25]. It is divided into three modules: a household roster, a breastfeeding initiation module (BF), and an infant and young child feeding (IYCF) module. Modified versions of this questionnaire have been used in several breastfeeding studies [16–19] and have adapted the standard WHO "recall since birth" method for local cultural contexts, local practices, and also to confine it within the limits of feeding infants less than the age of six months. EBF was determined by maternal reports confirming no non-breast milk foods/liquids since birth or in the prior 24 hours. This aligned closely with WHO indicators (no liquids/solids beyond breast milk, medicines, or oral rehydration up to the reference age, 6 months), though adaptations addressed local practices.

To minimize social desirability bias, several interviewer training and data collection strategies were implemented. Interviewers received standardized training emphasizing rapport building, trust, and neutrality during interactions. They were instructed to maintain neutral facial expressions and tone, avoid evaluative comments such as praise, and read questions

verbatim without implying normative responses. Training also covered techniques to normalize potentially sensitive behaviors such as "Many mothers say they sometimes..." and to begin interviews with light, non-threatening conversations to build rapport before addressing sensitive questions. Participants were assured verbally and in the written consent form that their decision to participate and their responses would not influence the care they received at the clinic. In addition, more sensitive questions were placed later in the questionnaire after trust was established.

### Sample size and sampling procedure

Sampling was conducted using a convenience sampling approach. Eligible mother–infant pairs were recruited from high-volume postnatal care facilities on the days of data collection. The sample size was therefore determined by the number of mothers attending the clinics and consenting to participate during the recruitment period. A total of 130 mother–infant pairs were enrolled, of whom 102 met the inclusion criteria, with infants aged between 6 weeks and 5.5 months. Categorization of infants into age groups was performed during data analysis. Data on maternal socioeconomic characteristics such as age, education level, marital status, and employment status were collected using a structured questionnaire administered during postnatal care visits.

### Statistical analysis

Quantitative data were analyzed using Stata version 13.1 (StataCorp LLC, College Station, TX, USA, 2013). The dataset was systematically organized into thematic categories (e.g., mother and baby credentials), reviewed, and coded. For analytical purposes, a total of 102 infants were categorized into three age groups for data analysis: 6 weeks to 2.4 months (n = 26), 2.5–4.0 months (n = 43); and 4.1–5.5 months (n = 33). To assess the validity of maternal self-reported infant feeding practices of EBF compared to those objectively classified using the deuterium oxide DTM method, a Kappa (κ) analysis was conducted. Kappa statistics evaluate the agreement between two classification methods beyond what would be expected by chance alone. The κ coefficient ranges from −1.0 to 1.0, where 1.0 represent a perfect agreement, 0.0 indicates agreement equivalent to chance, and −1.0 representing a perfect disagreement. Further classifications are: 0.01–0.20 slight agreement; 0.21–0.40 fair agreement; 0.41–0.60 moderate agreement; 0.61–0.80 substantial agreement, and 0.81–0.99 almost perfect agreement [26].

## Results

### Study population

Of the 130 mother-baby-pairs recruited, 102 met the inclusion criteria, with infants aged between 6 weeks and 5.5 months. Maternal age ranged from 16 to 49 years. The majority of mothers reported some or completed high school as the highest level of education (74.5%), were unemployed (86.3%) or single (56.9%), as shown in Table 1.

### Breastfeeding categories using the deuterium-oxide DTM method for model 1 and model 2

Using the more stringent Model 1 (NMOI ≤ 25 g/day), EBF prevalences were 50% among infants aged ≤2.4 months, 18.2% in the 2.5–4.0-month group, and 18.8% among those aged 4.1–5.5 months, as shown in Fig 1. I n contrast, applying the less stringent Model 2 (NMOI ≤ 86.6 g/day) resulted in higher EBF prevalences of 65.4%, 50%, and 43.8%, respectively, across the same age groups. Partial breastfeeding remained consistent across both models, accounting for 3.8% in the ≤ 2.4-month group, 20.5% in the 2.5–4.0-month group, and 43.8% in the 4.1–5.5-month group.

### Comparison of EBF prevalences between DTM methods and maternal recall

Among infants aged ≤2.4 months, maternal recall indicated an EBF prevalence of 100%, as shown in Fig 2. In contrast, when assessed objectively using the DTM method, the EBF prevalence was substantially lower: 50% based on Model 1

**Table 1. Socio-demographic characteristics of the mother-infant pairs in the study.**

| Maternal Characteristics n = 102 | N | % |
|---|---|---|
| **Maternal age (years)** | | |
| 16-20 | 19 | 18.6 |
| 21-25 | 37 | 36.3 |
| 26-30 | 16 | 15.7 |
| 31-35 | 17 | 16.7 |
| 36-49 | 13 | 12.7 |
| **Age of infants (months)** | | |
| ≤2.4 | 26 | 25.5 |
| 2.5-4 | 32 | 31.4 |
| 4.1-5.5 | 44 | 43.1 |
| **Education level** | | |
| Primary | 14 | 13.7 |
| Secondary | 76 | 74.5 |
| Tertiary | 12 | 11.8 |
| **Marital status** | | |
| Married | 41 | 40.2 |
| Cohabiting | 3 | 2.9 |
| Single | 58 | 56.9 |
| **Employment status** | | |
| Employed | 12 | 11.8 |
| Self employed | 2 | 2.0 |
| Unemployed | 88 | 86.3 |

(NMOI ≤ 25 g/day) and 65.4% based on Model 2 (NMOI ≤ 86.6 g/day). Among infants aged 2.5–4.0 months, EBF prevalences decreased further, with maternal recall reporting 80%, whereas DTM-based estimates showed only 22.7% using Model 1 and 50% using Model 2. Similarly, in the 4.1–5.5-month age group, maternal recall indicated an EBF prevalence of 70%, but DTM assessment revealed markedly lower values—approximately 21.9% using Model 1 and 43.8% based on Model 2.

The Kappa analysis, which compared EBF by maternal recall to objectively determined EBF using Models 1 and 2 gave Kappa values of −0.25 and −0.27 respectively, as shown in Table 2. These Kappa values represent a fair disagreement.

## Discussion

Our study assessed EBF practices among breastfeeding mothers in Eswatini, a country with a persistently high child malnutrition rate. To the best of our knowledge, this is the first study in Eswatini to investigate exclusive breastfeeding using the DTM method. EBF remains an important factor for the wellbeing of both infants and children [24]. Assessment of EBF was determined by maternal recall (subjective measure) and the DTM stable isotope method (an objective measure). The latter was assessed using two different cutoff points – the less stringent one, Model 2, which allows for a higher non-breast milk intake (NMOI ≤86.6 g/day) and the more stringent Model 1 (NMOI ≤25 g/day). Measuring EBF has been explored extensively in both past and recent scholarly work [16,18,27–31]. Recent advancements have improved the DTM technique, enabling more accurate breastfeeding assessments through enhanced data collection, management, and validation processes [23,32].

In this study, most participants had only secondary education (74.5%) and were unemployed (86.3%). These socio-demographic characteristics may have contributed to the low prevalence of exclusive breastfeeding observed, as limited education and unemployment can restrict access to accurate breastfeeding information, reduce engagement with maternal health services, and affect women's confidence or autonomy in infant feeding decisions. Similar patterns have been

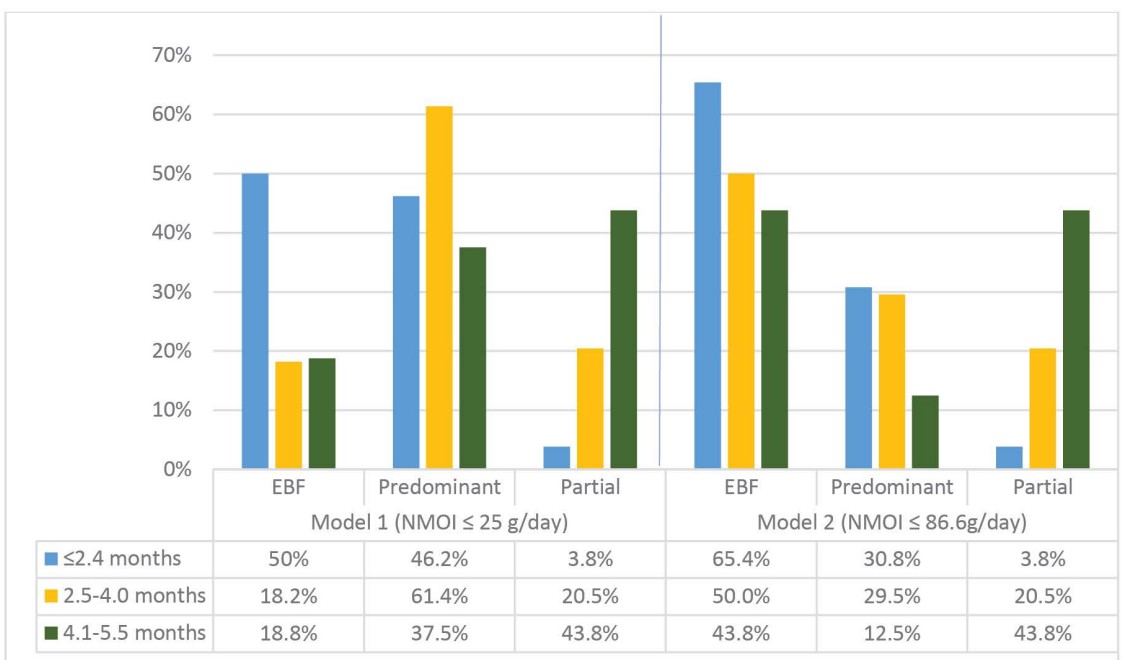

**Fig 1. Comparison of breastfeeding categories using the deuterium-oxide dose-to-mother method for Model 1 and 2 by age.** NMOI, non-milk oral intake.

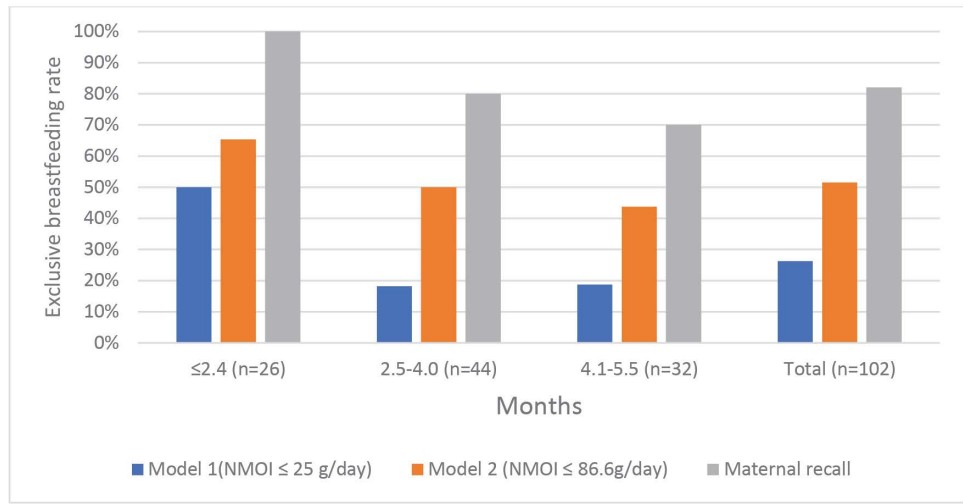

**Fig 2. Comparison of exclusive breastfeeding prevalences between the dose-to-mother method and maternal recall.** NMOI, non-milk oral intake.

reported in studies from East and Southeast Asia, where lower education and unemployment were significantly associated with reduced exclusive breastfeeding prevalences [33,34].

Among infants aged ≤2.4 months, maternal recall indicated an EBF prevalence of 100% which was in contrast to DTM method results of 50% based on Model 1 and 65.4% based on Model 2. These findings showed that some infants were

**Table 2. Kappa analysis of exclusive breastfeeding from maternal recall vs the objective dose-to-mother method using two different cuts off for NMOI (Model 1 and 2).**

| Model 1 (NMOI ≤ 25 g/day) | | | | | |
|---|---|---|---|---|---|
| Agreement | Expected Agreement | Kappa | Standard Error | Z | Prob>Z |
| 49.02% | 59.23% | −0.2504 | 0.0986 | −2.54 | 0.9945 |
| **Model 2** (NMOI ≤ 86.6g/day) | | | | | |
| Agreement | Expected Agreement | Kappa | Standard Error | Z | Prob>Z |
| 35.29% | 49.23% | −0.2745 | 0.0896 | −3.06 | 0.9989 |

NMOI, non-milk oral intake.

introduced to non-milk oral intake (NMOI), which is water intake from water, other milks, beverages, supplementary foods, or their combinations, before the age of 3 months, indicating poor adherence to the WHO recommendation of EBF for the first 6 months [6]. These results concur with studies conducted in Botswana, India, Cameroon, and Malaysia [17,18, 35,36]. In Botswana all mothers reported practicing EBF according to the maternal recall method, however only 61.2% were actually doing so based on the DTM method at three months. A similar discrepancy was also observed in India by Samuel *et al.* [36], where 90% of mothers reported EBF via maternal recall, but only 23% were confirmed EBF using the DTM method. In addition, a study in Cameroon reported that about 75% of the mothers reported that they were exclusively breastfeeding their infants yet the DTM technique detected that only 11% of the infants were actually exclusively breastfed. Finally, in Malaysia all mothers stated that they were not giving their infants any other food or drinks except breast milk but the DTM method revealed that only 3% of these mothers were exclusively breastfeeding their infants [35].

The United Nations Children's Fund (UNICEF) has consistently reported the negative consequences of poor EBF on infant and child development including stunting and poor general wellbeing [8]. It is of great concern, therefore, that in the current study, the results from maternal recall overestimated EBF prevalences when compared to the DTM method. Currently policy interventions are based on maternal recall and may not have been effective in ensuring optimal infant feeding practices. The Kappa analysis further confirms that there was only fair disagreement between the mother's report of EBF and that obtained using the DTM method for both the Model 1 and Model 2 cutoff values for NMOI. The lack of correlation could suggest that mothers may be reporting breastfeeding practices in a way they think is socially desirable or expected, even if their actual practices are different, the so-called social desirability bias [37]. Despite efforts to prevent social desirability bias, its complete avoidance was unlikely because data collection occurred within a health facility that regularly provided participants with training and counseling on the advantages of exclusive breastfeeding. In the study by Stewart et al. [38], social desirability bias significantly influenced self-reported EBF outcomes in a Kenyan randomized trial promoting infant feeding practices. Researchers observed that mothers in the nutrition intervention group, exposed to breastfeeding promotion, frequently altered their reports over repeated surveys—75.9% shifted from reporting EBF cessation before 6 months to 6 months or later, compared to only 32.5% in the non-nutrition group—suggesting recall bias rather than true behavioral change. This highlights the need for objective EBF measures, like deuterium dose-to-mother techniques, to mitigate such reporting distortions in intervention studies. This compromises the information required to implement measures to improve child nutrition.

The study strengths included the ability to apply both objective and subjective measures to assess EBF practices in Eswatini. Secondly, close collaboration with national and facility-level stakeholders throughout site selection, implementation and feedback of results is likely to have increased the relevance and applicability of the findings for Eswatini's breastfeeding and child-nutrition programs, although it did not affect the objective measurement of EBF itself. However, the DTM method applied is not without limitations. Even though it accurately measures breast milk and NMOI over a 14-day observation period-allowing for an objective assessment of EBF, it may not reflect breastfeeding practices beyond that timeframe. Breastfeeding patterns can vary over time, and a mother may switch from mixed feeding to EBF when

provided with appropriate support. Similarly, this limitation applies to the 24-hour maternal recall method, which may not reliably reflect feeding behavior over extended time periods [19].

## Conclusions

This study validates the deuterium oxide dose-to-mother (DTM) technique as the biochemical gold standard for assessing exclusive breastfeeding (EBF) against maternal recall in Eswatini. Maternal recall substantially overestimated EBF prevalence across infant age groups, revealing inconsistencies due to recall inaccuracies and social desirability bias. These findings confirm DTM's superiority for validation studies, despite measuring breastfeeding over a short observation period. Integrating DTM enhances methodological rigor in breastfeeding research.

## Acknowledgments

We thank the Ministry of Health of Eswatini for their technical and logistical support, and for granting us permission to implement this study. Our appreciation extends to the Eswatini Health Laboratory for providing sample storage and analysis. We sincerely acknowledge the management of the health facilities where data was collected for their cooperation. Finally, the authors wish to acknowledge the mothers and infants who participated in the study and the research assistants who assisted with questionnaires and saliva sampling.

## Author contributions

**Conceptualization:** Kwanele Siyabonga Simelane.

**Data curation:** Kwanele Siyabonga Simelane, Nhlakanipho Sandziso Mkhaliphi, Glorious Dlamini, Helen Mulol, Anna Coutsoudis.

**Formal analysis:** Nhlakanipho Sandziso Mkhaliphi, Anna Coutsoudis.

**Investigation:** Kwanele Siyabonga Simelane, Tholakele Mhlanga, Siniketiwe Zwane.

**Methodology:** Kwanele Siyabonga Simelane, Tholakele Mhlanga, Siniketiwe Zwane, Glorious Dlamini, Henry Gadaga, Helen Mulol, Anna Coutsoudis.

**Supervision:** Tholakele Mhlanga, Siniketiwe Zwane, Glorious Dlamini, Henry Gadaga.

**Validation:** Henry Gadaga, Helen Mulol, Anna Coutsoudis.

**Writing – original draft:** Kwanele Siyabonga Simelane.

**Writing – review & editing:** Kwanele Siyabonga Simelane, Tholakele Mhlanga, Nhlakanipho Sandziso Mkhaliphi, Siniketiwe Zwane, Glorious Dlamini, Henry Gadaga, Helen Mulol, Anna Coutsoudis.

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
