## [Decision Letter · Decision Letter 0]

2 Jan 2026

PONE-D-25-44019Validating self-reported exclusive breastfeeding in Eswatini using stable isotope techniquesPLOS One

Dear Dr. Simelane,

Thank you for submitting your manuscript to PLOS ONE. After careful consideration, we feel that it has merit but does not fully meet PLOS ONE’s publication criteria as it currently stands. Therefore, we invite you to submit a revised version of the manuscript that addresses the points raised during the review process.

We look forward to receiving your revised manuscript.

Kind regards,

Jennifer Yourkavitch

Academic Editor

PLOS One

Journal Requirements:

2. You have indicated that data is available from “ehhrrbeswatini@gmail.com.”  Please can we ask you to provide us with a general contact email address for the data requests, so readers can request access in perpetuity. If a general email is not available please provide a link to a website where readers can obtain access to data.

Additional Editor Comments:

Please respond to each point from both reviewers.

Reviewers' comments:

Reviewer's Responses to Questions

**Comments to the Author**

1. Is the manuscript technically sound, and do the data support the conclusions?

Reviewer #1: Partly

Reviewer #2: Partly

2. Has the statistical analysis been performed appropriately and rigorously?

Reviewer #1: I Don't Know

Reviewer #2: Yes

3. Have the authors made all data underlying the findings in their manuscript fully available?

Reviewer #1: No

Reviewer #2: Yes

4. Is the manuscript presented in an intelligible fashion and written in standard English?

Reviewer #1: Yes

Reviewer #2: Yes

5. Review Comments to the Author

Reviewer #1: I think the paper presents original research that has not been conducted in this setting and contributes to a growing body of research on the methods used to measure exclusive breastfeeding. However, some parts of the paper provided insufficient detail to assess the quality of the analysis. Furthermore, the conclusion is not supported by the data presented and I recommend this section be rewritten. Line by line comments below:

Line 19: the EBF data collected using the 24-hour recall presented is a prevalence (i.e. a snapshot of the percent of infants who are exclusively breastfed at one point in time), not a rate. Suggest reframing throughout the paper as the "prevalence of exclusive breastfeeding".

Lines 31-34: In the abstract, referring to the models as Model 1 and Model 2 is not sufficient for readers to understand the difference. Suggest clarifying that the difference between the two models is the cutoff that was used to determine EBF using the DTM.

Lines 36-38: The conclusion to strengthen breastfeeding education does not flow logically from the data presented. For one, this is a recommendation, not a conclusion. The data presented suggests that there is a discrepancy between the DTM and the recall method, most likely due to overreporting. In my opinion, if a recommendation were to emerge from this finding, it would be related to improving the measurement of EBF or recommendations about interpreting 24 hour recall EBF data when using it for program monitoring and evaluation.

Line 44-46: There is evidence that stunting that occurs before age 2 is potentially reversible under the right conditions. So I checked the reference cited here. This source does not seem to support the sentence. Please cite another source, or rewrite to align with the the data in this source.

Line 47: Define EBF here.

Line 49: Stunting is not a risk factor for poor child development per se, but a marker of poor child development (see LeRoy and Frongillo 2019).

Lines 51-53: Since this manuscript is solely about EBF, suggest removing references to other IYCF monitoring.

Lines 53-54: Since EBF here refers to the prevalence estimate, suggest revising to read "EBF prevalence among 0-6 month olds" to be more precise.

Lines 68: The manuscript did not mention any policies in Eswatini nor how these findings could be used to inform or change policies. The data presented in this manuscript seems to assess the validity of EBF measured using the recall method; or to improve understanding of differences in EBF measurement by comparing recall and DTM. The purpose/goal of the study should be restated.

Line 74: Please clarify whether this is the national prevalence of poverty or whether this is referring to rural poverty. It is unclear as stated. Also clarify whether the 54.8% refers to people or households.

Lines71-79: Suggest moving to background section.

Lines 102-130: This section combines the sample collection and processing with the analysis of the samples. If the editor agrees, suggest separating these two and describing the laboratory analysis of the saliva sample in the analysis section.

Lines 134-135: The authors state that, "Model 2 was developed based on 24-hour observations of mothers practicing EBF." However, the study methods do not include observations, so this is confusing. I believe Model 2 only differs from Model 1 in that different cutoffs for NMOI were used to determine exclusive, predominant or partial BF. Also, please add a reference and clarify where the 86.6 cutoff came from.

Lines 138-145: The description of the DTM sample collection and analysis is very detailed. However, there are few details about how the 24 hour recall data were collected. Where were mothers interviewed? When were they interviewed (at the beginning or at the end of the 14 day period)? How was EBF determined? Was it according to the WHO guidance, exactly? Authors state that the WHO recommended questionnaire was modified, but do not say how, which makes me wonder whether the analysis was modified as well.

Lines 156-158: This needs a reference.

Line 163: Please clarify what "up to high school education" means. Does this mean that 74.5% reported high school as the highest level of education completed? Does it mean 74.5% reported at least some high school education? Does this mean that 74.5% completed high school or less?

Figure 1, Model 1: I am surprised to see that EBF was the same in the middle and older age group. I would have expected more of a difference. Have other studies had similar findings?

Line 198: On line 156, the authors state that a score of -1.0 is "perfect disagreement" and that 0.21-0.40 [positive] represent fair agreement. So, I am confused that these negative kappa coefficients are described as "fair agreement" instead of fair disagreement. Please reconcile.

Lines 213-216: This paragraph does not seem very relevant. Suggest deleting or revise to make the relevance to your findings more explicit.

Lines 219-220: Do the methods allow you to distinguish between consumption of supplementary foods, other milks, beverages, and water? Could the non breast milk consumption be any of these things? I am not sure why it only mentions foods. Unless data from the survey suggests it is food and not water or other milks.

Line 240: Social desirability bias likely results in the overestimation. However, it was difficult to assess how much it might have played a roll here. For example, the mother-infant dyads were recruited from health centers, so presumably all of them had been exposed to breastfeeding promotion messages. This may increase this bias. Other factors like who interviewed them and when are not mentioned, making it difficult to assess. Finally, were any steps taken to reduce social desirability bias? For example, introductory language before the questions, asking questions in different ways, etc. There is not enough detail in the methods section about how the questionnaire was administered to get a good grasp of this.

Lines 243-252: It is worth mentioning that the 24 hour recall is just one day of the 14 observed using the DTM (assuming the interview occurred within the 14 day period, which is not stated in the methods). So complete agreement is not expected if mothers do not feed their infants in the same way every single day. It is also worth discussing how the 24 hour method is a proxy, not a direct measure, of EBF for the duration of the first six months as recommended by WHO, though it is often misinterpreted as such.

Lines 254-260: The conclusions are not supported by the findings. Suggest rewriting this section completely.

Reviewer #2: Thanks for the opportunity to review this paper, which compares EBF as reported using the WHO IYCF recall method and a stable isotope technique. It is helpful to have objective measures of EBF to better understand the reliability of the recall method. I think this is an important paper, but my overarching comment is about the framing. It’s not clear why this needs to be framed as important for child stunting. There are many other and more well documented reasons EBF is important for the health of infants. I would suggest framing the paper (in the intro and discussion) so that it focuses on the importance of accurate measurement of EBF and the potential influence of social desirability on survey responses. The discussion could talk more about what low agreement between the recall and isotope methods means for policy and practice. Countries are making policies based on the prevalence of EBF from Demographic and Health Surveys, Multiple Indicator Cluster Surveys, or SMART surveys. What should they do now, if the survey data on EBF are not accurate?

In addition to my overarching comments, I would like to offer the following specific comments to the authors:

Lines 44-45 – I suggest either removing the word “irreversible” or switching the order of “irreversible stunting” and “acute malnutrition.” The current order makes it seem like acute malnutrition is also irreversible, which is not true.

Lines 48-50 – There is not much evidence that EBF is a protective factor against stunting. The Victora et al. paper cited does not show that EBF is related to stunting reduction. I suggest revising this sentence to highlight the documented benefits of EBF (as outlined in the Victora paper).

Lines 50-52 – Please revise this sentence. IYCF indicators are not used to monitor malnutrition trends. Growth indicators (WHZ, HAZ, WAZ) are used for monitoring malnutrition trends.

First paragraph of Background section – It’s not clear why the authors are emphasizing the connection between EBF and stunting in the first paragraph of the Background section. It seems like it would make sense to focus in this paragraph on the well-established benefits of EBF.

Line 92 – How did you determine if the mothers were “clinically well”?

Lines 139-140 – Please explain what kind of adaptations were made to the WHO IYCF questionnaire.

Methods – Please include a sentence in the methods section stating that you collected data on the mothers’ socioeconomic characteristics. Please add a subsection on your sample size calculation and how you decided on the number of infants to include in each age group. In addition, it would be helpful to describe the sampling process. Was it purely convenience sampling or did you do stratified sampling by infant age?

Table 1 – What do the numbers at the far-right side of the table represent? If they are not important, I would suggest removing them.

Section starting on line 188 – Please add sentences to this section for the other two age groups. The text in this section currently only describes results for the youngest age group.

Table 1 – I suggest using the same number of decimal places throughout the table. Two decimal places should be sufficient.

Line 204 – Please be more specific with your statement here by including what the first study is about.

Lines 209-210 – It’s not clear what issue related to measuring EBF the authors are referring to here. Please add more information about the issues mentioned.

Lines 213-216 – It’s not clear how this is relevant to your main findings. Depending on the context, women with higher levels of education are less likely to EBF than women with lower levels. So, is your point that these women are not well enough educated or too well educated?

Line 236 – I would suggest replacing “addressing child malnutrition” with “ensuring optimal infant feeding practices.”

Line 241 – You may want to add a sentence about social desirability in the measurement of EBF in relation to this study by Christine Stewart et al., 2025: https://www.sciencedirect.com/science/article/pii/S247529912401713X

Line 244-246 - This is the first time a stakeholder participatory approach is mentioned. If you are going to talk about this as a strength, please mention it in the methods section, although I don’t understand how a participatory approach is relevant as a strength in this case.

Lines 255-256 – I suggest removing the sentence “Hence there is a need to strengthen breastfeeding education among mothers to reduce the rate of childhood malnutrition in Eswatini.” This conclusion does not follow from your results. In fact, it is likely that the breastfeeding education women are already receiving is contributing to the socially desirable responses they are giving on the WHO IYCF questionnaire.

6. PLOS authors have the option to publish the peer review history of their article (what does this mean?). If published, this will include your full peer review and any attached files.

Reviewer #1: No

Reviewer #2: No

---

## [Author Response · Author response to Decision Letter 1]

4 Mar 2026

General Comment.

You have indicated that data is available from “ehhrrbeswatini@gmail.com.” Please can we ask you to provide us with a general contact email address for the data requests, so readers can request access in perpetuity. If a general email is not available, please provide a link to a website where readers can obtain access to data.

Response

The general contact email address for data requests, through which readers can obtain access on an ongoing basis, is “es@ehhrrb.org.sz”. This address is managed by the Eswatini Health and Human Research Review Board (EHHRRB), the national health research ethics committee under the Eswatini Ministry of Health. It is actively monitored by the Board’s administrator.

Responses to Reviewer’s Comments

Reviewer #1

1. Comment

Line 19: the EBF data collected using the 24-hour recall presented is a prevalence (i.e. a snapshot of the percent of infants who are exclusively breastfed at one point in time), not a rate. Suggest reframing throughout the paper as the "prevalence of exclusive breastfeeding".

Response

Thank you for your comment. We have changed this throughout the paper as suggested. Example below:

Original Text

Exclusive breastfeeding (EBF) rates at the population level are typically assessed through maternal or caregiver reports, which are prone to recall and social desirability biases.

Revised text

Exclusive breastfeeding (EBF) prevalence at the population level is typically assessed through maternal or caregiver reports, which are prone to recall and social desirability biases.

2. Comment

Lines 31-34: In the abstract, referring to the models as Model 1 and Model 2 is not sufficient for readers to understand the difference. Suggest clarifying that the difference between the two models is the cutoff that was used to determine EBF using the DTM.

Response

We appreciate the valuable comment. Clarity has been provided.

Original Text

Using Model 1 cutoffs, EBF rates by maternal recall vs. DTM were 100% vs. 50% (≤2.4 months), 79.5% vs. 18.2% (2.5–4 months), and 71.9% vs. 18.8% (4.1–5.5 months). For Model 2, rates were 100% vs. 65.4%, 79.5% vs. 50%, and 71.9% vs. 43.8%, respectively.

Revised text

Using Model 1 cutoffs, prevalences of EBF by maternal recall vs. DTM were 100% vs. 50% (≤2.4 months), 79.5% vs. 18.2% (2.5–4 months), and 71.9% vs. 18.8% (4.1–5.5 months). For Model 2, rates were 100% vs. 65.4%, 79.5% vs. 50%, and 71.9% vs. 43.8%, respectively. The difference between the two models is the cutoff that was used to determine EBF using the DTM method.

3. Comment;

Lines 36-38: The conclusion to strengthen breastfeeding education does not flow logically from the data presented. For one, this is a recommendation, not a conclusion. The data presented suggests that there is a discrepancy between the DTM and the recall method, most likely due to overreporting. In my opinion, if a recommendation were to emerge from this finding, it would be related to improving the measurement of EBF or recommendations about interpreting 24 hour recall EBF data when using it for program monitoring and evaluation.

Response

Thank you for your valuable comments, the conclusion has been revised to align with the findings.

Original text

The lower EBF rates detected by DTM highlight the urgent need to strengthen breastfeeding education in Eswatini to combat childhood malnutrition.

Revised text

Maternal recall substantially overestimated exclusive breastfeeding prevalence, while the deuterium oxide technique yields more reliable, objective assessments of infant feeding practices, highlighting the need to interpret recall data with caution due to overreporting.

4. Comment

Line 44-46: There is evidence that stunting that occurs before age 2 is potentially reversible under the right conditions. So I checked the reference cited here. This source does not seem to support the sentence. Please cite another source, or rewrite to align with the data in this source.

Response

Thank you for your comment. The source has been revised and the wording of the sentence has been arranged such that “acute malnutrition” comes before the “irreversible stunting”.

Original Text

The first two years of life represent a critical window for fostering healthy growth, establishing positive behaviours, and preventing irreversible stunting and acute malnutrition, ensuring that children reach their full developmental potential (2).

Revised text

The first two years of life represent a critical window for fostering healthy growth, establishing positive behaviours, and preventing acute malnutrition and stunting, ensuring that children reach their full developmental potential (5).

5. Comment

Line 47: Define EBF here.

Response

Thank you. Exclusive breastfeeding has been defined.

Original text

The World Health Organization (WHO) recommends that babies be exclusively breastfed for the first six months after which complementary feeding must be introduced (3)

Revised text

The World Health Organization (WHO) recommends that babies be exclusively breastfed (i.e., an infant receives only breast milk, with no other liquids or solids, not even water, for the first six months of life, except for oral rehydration solution, vitamin/mineral drops, or medicines) after which complementary feeding must be introduced (6).

6. Comment: Line 49: Stunting is not a risk factor for poor child development per se, but a marker of poor child development (see LeRoy and Frongillo 2019).

Response

Thank you for the valuable comment, indeed stunting is not necessarily a risk factor but is a marker of poor child development. Therefore, the wording of the sentence has been revised.

Original text

Exclusive breastfeeding (EBF) is a major protective factor against stunting, which is a well-established risk for poor child development (4–6).

Revised text

Exclusive breastfeeding (EBF) is a major protective factor against stunting, which is a well-established marker for poor child development (7-12).

7. Comment

Lines 51-53: Since this manuscript is solely about EBF, suggest removing references to other IYCF monitoring.

Response

IYCF monitoring information has been removed.

Original text

It is therefore imperative for countries to routinely collect and interpret Infant and Young Child Feeding (IYCF) data to monitor malnutrition trends toward country specific and global targets and develop evidence-based policies and programs to address feeding practices. EBF is one of critical data elements for IYCF.

Revised text

It is therefore imperative for countries to routinely collect and interpret infant feeding and growth indicator data to monitor malnutrition trends toward country specific and global targets and develop evidence-based policies and programs to address feeding practices. EBF is one of the critical data elements for infant feeding.

8. Comment

Lines 53-54: Since EBF here refers to the prevalence estimate, suggest revising to read "EBF prevalence among 0-6 month olds" to be more precise.

Response

Thank you. We agree with your suggestion, and the wording has been revised.

Original text

As of 2022 in Eswatini, EBF rates during the first six months measured through maternal recall assessments were reported at 77.28%, while stunting among children under five years remained significant at 25% (7,8).

Revised text

As of 2022 in Eswatini, the prevalence of EBF 0-6 months old, measured through maternal recall assessments, was reported at 77.28%, while stunting among children under five years remained significant at 25% (13,14).

9. Comment

Lines 68: The manuscript did not mention any policies in Eswatini nor how these findings could be used to inform or change policies. The data presented in this manuscript seems to assess the validity of EBF measured using the recall method; or to improve understanding of differences in EBF measurement by comparing recall and DTM. The purpose/goal of the study should be restated.

Response

Thank you very much for this valuable comment. We have restated the study’s purpose to clarify that the main objective was to assess the validity of EBF measurement methods and to provide evidence that could inform improvements in national monitoring and policy strategies.

Original text

The goal is to generate more accurate data on breastfeeding practices to inform policy.

Revised text

The main objective is to assess the validity of EBF measurement methods and to provide evidence that could inform improvements in national monitoring and policy strategies.

10. Comments

Line 74: Please clarify whether this is the national prevalence of poverty or whether this is referring to rural poverty. It is unclear as stated. Also clarify whether the 54.8% refers to people or households.

Response

The World Bank “poverty headcount ratio at national poverty lines (% of population)” for Eswatini is defined as the percentage of the population living below the national poverty line. World Bank poverty headcount indicators are consistently person‑based (population share), even though they are calculated from household survey data. We have revised the text to address your questions.

Original Text

About 75.6% of the population live in rural areas (15), with a poverty rate of 54.8% (16).

Revised text

Nationally, 54.8% of the population lives below the national poverty line, with poverty concentrated in rural areas where about 70% of rural residents are poor (21). Approximately 75.6% of the population resides in rural areas.

11. Comment

Lines71-79: Suggest moving to background section.

Response

The text has been moved as suggested.

12. Comment

Lines 102-130: This section combines the sample collection and processing with the analysis of the samples. If the editor agrees, suggest separating these two and describing the laboratory analysis of the saliva sample in the analysis section.

Response

Thank you for your comment. Sample collection is now separated from sample analysis.

13. Comment

Lines 134-135: The authors state that, "Model 2 was developed based on 24-hour observations of mothers practicing EBF." However, the study methods do not include observations, so this is confusing. I believe Model 2 only differs from Model 1 in that different cutoffs for NMOI were used to determine exclusive, predominant or partial BF. Also, please add a reference and clarify where the 86.6 cutoff came from.

Response

Thank you for your comment. We agree that our original statement could be misinterpreted. Model 2 does not involve direct observations from our dataset; rather, the NMOI cutoff applied in this model was derived from a previous study (Liu et al., 2019) that included observed mother–infant pairs practicing exclusive breastfeeding in Indonesia. This cutoff (86.6) corresponds to the 90th percentile of the posterior distribution of NMOI in that population. We have revised the text to clarify this and added the Liu et al. study as a new reference (now reference 23). Model 2 differs from Model 1 only in the use of this empirically derived cutoff to classify exclusive breastfeeding status.

Reference added:

23. Liu Z, Diana A, Slater C, Preston T, Gibson RS, Houghton L, Duffull SB. Development of a nonlinear hierarchical model to describe the disposition of deuterium in mother–infant pairs to assess exclusive breastfeeding practice. J Pharmacokinet Pharmacodyn. 2019;46(1):1–13. https://doi.org/10.1007/s10928-018-9613-xComment

14. Lines 138-145: The description of the DTM sample collection and analysis is very detailed. However, there are few details about how the 24 hour recall data were collected. Where were mothers interviewed? When were they interviewed (at the beginning or at the end of the 14 day period)? How was EBF determined? Was it according to the WHO guidance, exactly? Authors state that the WHO recommended questionnaire was modified, but do not say how, which makes me wonder whether the analysis was modified as well.

Response

Thank you for your valuable comment. We have revised this section to clear all the grey areas. Below is the revised paragraph.

Original Text

Maternal recall was assessed using a questionnaire on infant feeding practices. Breastfeeding indicators were obtained through interviews with mothers, using a questionnaire adapted from the WHO/UNICEF Indicators for Assessing Infant and Young Child Feeding, Part 2 (18). The tool is based on current guidelines regarding breastfeeding, complementary feeding, and the feeding of non-breastfed infants and young children under two years of age (18). It is divided into three modules: a household roster, a breastfeeding initiation module (BF), and an infant and young child feeding (IYCF) module. Modified versions of this questionnaire have been used in several breastfeeding studies (10–13).

Revised text

Maternal recall was assessed using a questionnaire on infant feeding practices. Through an interview, mothers reported infant feeding practices over the previous 24 hours, detailing breast milk intake alongside any liquids, semi-solids, or solids given. Breastfeeding indicators were obtained through interviews with mothers, using a questionnaire adapted from the WHO/UNICEF Indicators for Assessing Infant and Young Child Feeding, Part 2 (24) at the onset of sample collection, just before participants received health services at the facility. The tool is based on current guidelines regarding breastfeeding, complementary feeding, and the feeding of non-breastfed infants and young children under two years of age (25). It is divided into three modules: a household roster, a breastfeeding initiation module (BF), and an infant and young child feeding (IYCF) module. Modified versions of this questionnaire have been used in several breastfeeding studies (16–19) and have adapted the standard WHO "recall since birth" method for local cultural contexts, local practices, and also to confine it within the limits of feeding infants less than the age of six months. EBF was determined by maternal reports confirming no non-breast milk foods/liquids since birth or in the prior 24 hours. This aligned closely with WHO indicators (no liquids/solids beyond breast milk, medicines, or oral rehydration up to the reference age, 6 months), though adaptations addressed local practices.

15. Comment

Lines 156-158: This needs a reference.

Response; We have provided a reference.

Reference cited: McHugh ML. Interrater reliability: the kappa statistic. Biochem Medica. 2012;22(3):276–82.

16. Comment

Line 163: Please clarify what "up to high school education" means. Does this mean that 74.5% reported high school as the highest level of education completed? Does it mean 74.5% reported at least some high school education? Does this mean that 74.5% completed high school or less?

Response

In this case up to high school education means reached and/or completed high school. Some reached but did not complete high school. Others reported that they have reached and completed high school. So, the 74.5% is some or completed high school combined into one group.

Original text

The majority of mothers had up to high school education (74.5%), were unemployed (86.3%) or single (56.9%), as shown in….

Revised text

The majority of mothers reported some or completed high school as the highest level of education (74.5%), were unemployed (86.3%) or single (56.9%), as shown in Table 1.

17. Comment

Figure 1, Model 1: I am surprised to see that EBF was the same in the middle and older age group. I would have expected more of a difference. Have other studies had similar findings?

Response

No, most studies report declining exclusive breastfeeding (EBF) prevalences as infants age from younger (e.g., 0-2 months) to middle/older groups (e.g., 4-6 months) within the 0-6 month period, contrary to the similar EBF prevalences observed in Model 1 (Figure 1). Declining EBF prevalences were, however, found for the less stringent cut-off, which was the more expected trend, and this further supports the conclusion that Model 2 more accurately reflects true EBF than Model 1.

18. Comment

Line 198: On line 156, the authors state that a score of -1.0 is "perfect disagreement" and that 0.21-0.40 [positive] rep

---

## [Editor Report · Decision Letter 1]

9 Mar 2026

PONE-D-25-44019R1Validating self-reported exclusive breastfeeding in Eswatini using stable isotope techniquesPLOS One

Dear Dr. Simelane,

Thank you for submitting your manuscript to PLOS ONE. After careful consideration, we feel that it has merit but does not fully meet PLOS ONE’s publication criteria as it currently stands. Therefore, we invite you to submit a revised version of the manuscript that addresses the points raised during the review process.

We look forward to receiving your revised manuscript.

Kind regards,

Jennifer Yourkavitch

Academic Editor

PLOS One

Journal Requirements:

**Additional Editor Comments:**

Thank you for your thorough and thoughtful responses to reviewers. There is one outstanding item--Reviewer 1, comment 21, states: "There is not enough detail in the methods section about how the questionnaire was

administered to get a good grasp of this." Meaning of social desirability bias. You provided a thorough response to the reviewer but did not address this comment in the text. Since social desirability bias was named the major culprit for the discrepancy between recall and DTM, it's worth providing more explanation in the Methods section as you did in your response to reviewers.

---

## [Author Response · Author response to Decision Letter 2]

10 Mar 2026

General Comment.

You have indicated that data is available from “ehhrrbeswatini@gmail.com.” Please can we ask you to provide us with a general contact email address for the data requests, so readers can request access in perpetuity. If a general email is not available, please provide a link to a website where readers can obtain access to data.

Response

The general contact email address for data requests, through which readers can obtain access on an ongoing basis, is “es@ehhrrb.org.sz”. This address is managed by the Eswatini Health and Human Research Review Board (EHHRRB), the national health research ethics committee under the Eswatini Ministry of Health. It is actively monitored by the Board’s administrator.

Responses to Reviewer’s Comments

Reviewer #1

1. Comment

Line 19: the EBF data collected using the 24-hour recall presented is a prevalence (i.e. a snapshot of the percent of infants who are exclusively breastfed at one point in time), not a rate. Suggest reframing throughout the paper as the "prevalence of exclusive breastfeeding".

Response

Thank you for your comment. We have changed this throughout the paper as suggested. Example below:

Original Text

Exclusive breastfeeding (EBF) rates at the population level are typically assessed through maternal or caregiver reports, which are prone to recall and social desirability biases.

Revised text

Exclusive breastfeeding (EBF) prevalence at the population level is typically assessed through maternal or caregiver reports, which are prone to recall and social desirability biases.

2. Comment

Lines 31-34: In the abstract, referring to the models as Model 1 and Model 2 is not sufficient for readers to understand the difference. Suggest clarifying that the difference between the two models is the cutoff that was used to determine EBF using the DTM.

Response

We appreciate the valuable comment. Clarity has been provided.

Original Text

Using Model 1 cutoffs, EBF rates by maternal recall vs. DTM were 100% vs. 50% (≤2.4 months), 79.5% vs. 18.2% (2.5–4 months), and 71.9% vs. 18.8% (4.1–5.5 months). For Model 2, rates were 100% vs. 65.4%, 79.5% vs. 50%, and 71.9% vs. 43.8%, respectively.

Revised text

Using Model 1 cutoffs, prevalences of EBF by maternal recall vs. DTM were 100% vs. 50% (≤2.4 months), 79.5% vs. 18.2% (2.5–4 months), and 71.9% vs. 18.8% (4.1–5.5 months). For Model 2, rates were 100% vs. 65.4%, 79.5% vs. 50%, and 71.9% vs. 43.8%, respectively. The difference between the two models is the cutoff that was used to determine EBF using the DTM method.

3. Comment;

Lines 36-38: The conclusion to strengthen breastfeeding education does not flow logically from the data presented. For one, this is a recommendation, not a conclusion. The data presented suggests that there is a discrepancy between the DTM and the recall method, most likely due to overreporting. In my opinion, if a recommendation were to emerge from this finding, it would be related to improving the measurement of EBF or recommendations about interpreting 24 hour recall EBF data when using it for program monitoring and evaluation.

Response

Thank you for your valuable comments, the conclusion has been revised to align with the findings.

Original text

The lower EBF rates detected by DTM highlight the urgent need to strengthen breastfeeding education in Eswatini to combat childhood malnutrition.

Revised text

Maternal recall substantially overestimated exclusive breastfeeding prevalence, while the deuterium oxide technique yields more reliable, objective assessments of infant feeding practices, highlighting the need to interpret recall data with caution due to overreporting.

4. Comment

Line 44-46: There is evidence that stunting that occurs before age 2 is potentially reversible under the right conditions. So I checked the reference cited here. This source does not seem to support the sentence. Please cite another source, or rewrite to align with the data in this source.

Response

Thank you for your comment. The source has been revised and the wording of the sentence has been arranged such that “acute malnutrition” comes before the “irreversible stunting”.

Original Text

The first two years of life represent a critical window for fostering healthy growth, establishing positive behaviours, and preventing irreversible stunting and acute malnutrition, ensuring that children reach their full developmental potential (2).

Revised text

The first two years of life represent a critical window for fostering healthy growth, establishing positive behaviours, and preventing acute malnutrition and stunting, ensuring that children reach their full developmental potential (5).

5. Comment

Line 47: Define EBF here.

Response

Thank you. Exclusive breastfeeding has been defined.

Original text

The World Health Organization (WHO) recommends that babies be exclusively breastfed for the first six months after which complementary feeding must be introduced (3)

Revised text

The World Health Organization (WHO) recommends that babies be exclusively breastfed (i.e., an infant receives only breast milk, with no other liquids or solids, not even water, for the first six months of life, except for oral rehydration solution, vitamin/mineral drops, or medicines) after which complementary feeding must be introduced (6).

6. Comment: Line 49: Stunting is not a risk factor for poor child development per se, but a marker of poor child development (see LeRoy and Frongillo 2019).

Response

Thank you for the valuable comment, indeed stunting is not necessarily a risk factor but is a marker of poor child development. Therefore, the wording of the sentence has been revised.

Original text

Exclusive breastfeeding (EBF) is a major protective factor against stunting, which is a well-established risk for poor child development (4–6).

Revised text

Exclusive breastfeeding (EBF) is a major protective factor against stunting, which is a well-established marker for poor child development (7-12).

7. Comment

Lines 51-53: Since this manuscript is solely about EBF, suggest removing references to other IYCF monitoring.

Respnse

IYCF monitoring information has been removed.

Original text

It is therefore imperative for countries to routinely collect and interpret Infant and Young Child Feeding (IYCF) data to monitor malnutrition trends toward country specific and global targets and develop evidence-based policies and programs to address feeding practices. EBF is one of critical data elements for IYCF.

Revised text

It is therefore imperative for countries to routinely collect and interpret infant feeding and growth indicator data to monitor malnutrition trends toward country specific and global targets and develop evidence-based policies and programs to address feeding practices. EBF is one of the critical data elements for infant feeding.

8. Comment

Lines 53-54: Since EBF here refers to the prevalence estimate, suggest revising to read "EBF prevalence among 0-6 month olds" to be more precise.

Response

Thank you. We agree with your suggestion, and the wording has been revised.

Original text

As of 2022 in Eswatini, EBF rates during the first six months measured through maternal recall assessments were reported at 77.28%, while stunting among children under five years remained significant at 25% (7,8).

Revised text

As of 2022 in Eswatini, the prevalence of EBF 0-6 months old, measured through maternal recall assessments, was reported at 77.28%, while stunting among children under five years remained significant at 25% (13,14).

9. Comment

Lines 68: The manuscript did not mention any policies in Eswatini nor how these findings could be used to inform or change policies. The data presented in this manuscript seems to assess the validity of EBF measured using the recall method; or to improve understanding of differences in EBF measurement by comparing recall and DTM. The purpose/goal of the study should be restated.

Response

Thank you very much for this valuable comment. We have restated the study’s purpose to clarify that the main objective was to assess the validity of EBF measurement methods and to provide evidence that could inform improvements in national monitoring and policy strategies.

Original text

The goal is to generate more accurate data on breastfeeding practices to inform policy.

Revised text

The main objective is to assess the validity of EBF measurement methods and to provide evidence that could inform improvements in national monitoring and policy strategies.

10. Comments

Line 74: Please clarify whether this is the national prevalence of poverty or whether this is referring to rural poverty. It is unclear as stated. Also clarify whether the 54.8% refers to people or households.

Response

The World Bank “poverty headcount ratio at national poverty lines (% of population)” for Eswatini is defined as the percentage of the population living below the national poverty line.

World Bank poverty headcount indicators are consistently person‑based (population share), even though they are calculated from household survey data. We have revised the text to address your questions.

Original Text

About 75.6% of the population live in rural areas (15), with a poverty rate of 54.8% (16).

Revised text

Nationally, 54.8% of the population lives below the national poverty line, with poverty concentrated in rural areas where about 70% of rural residents are poor (21). Approximately 75.6% of the population resides in rural areas.

11. Comment

Lines71-79: Suggest moving to background section.

Response

The text has been moved as suggested.

12. Comment

Lines 102-130: This section combines the sample collection and processing with the analysis of the samples. If the editor agrees, suggest separating these two and describing the laboratory analysis of the saliva sample in the analysis section.

Response

Thank you for your comment. Sample collection is now separated from sample analysis.

13. Comment

Lines 134-135: The authors state that, "Model 2 was developed based on 24-hour observations of mothers practicing EBF." However, the study methods do not include observations, so this is confusing. I believe Model 2 only differs from Model 1 in that different cutoffs for NMOI were used to determine exclusive, predominant or partial BF. Also, please add a reference and clarify where the 86.6 cutoff came from.

Response

Thank you for your comment. We agree that our original statement could be misinterpreted. Model 2 does not involve direct observations from our dataset; rather, the NMOI cutoff applied in this model was derived from a previous study (Liu et al., 2019) that included observed mother–infant pairs practicing exclusive breastfeeding in Indonesia. This cutoff (86.6) corresponds to the 90th percentile of the posterior distribution of NMOI in that population. We have revised the text to clarify this and added the Liu et al. study as a new reference (now reference 23). Model 2 differs from Model 1 only in the use of this empirically derived cutoff to classify exclusive breastfeeding status.

Reference added:

23. Liu Z, Diana A, Slater C, Preston T, Gibson RS, Houghton L, Duffull SB. Development of a nonlinear hierarchical model to describe the disposition of deuterium in mother–infant pairs to assess exclusive breastfeeding practice. J Pharmacokinet Pharmacodyn. 2019;46(1):1–13. https://doi.org/10.1007/s10928-018-9613-xComment

14. Lines 138-145: The description of the DTM sample collection and analysis is very detailed. However, there are few details about how the 24 hour recall data were collected. Where were mothers interviewed? When were they interviewed (at the beginning or at the end of the 14 day period)? How was EBF determined? Was it according to the WHO guidance, exactly? Authors state that the WHO recommended questionnaire was modified, but do not say how, which makes me wonder whether the analysis was modified as well.

Response

Thank you for your valuable comment. We have revised this section to clear all the grey areas. Below is the revised paragraph.

Original Text

Maternal recall was assessed using a questionnaire on infant feeding practices. Breastfeeding indicators were obtained through interviews with mothers, using a questionnaire adapted from the WHO/UNICEF Indicators for Assessing Infant and Young Child Feeding, Part 2 (18). The tool is based on current guidelines regarding breastfeeding, complementary feeding, and the feeding of non-breastfed infants and young children under two years of age (18). It is divided into three modules: a household roster, a breastfeeding initiation module (BF), and an infant and young child feeding (IYCF) module. Modified versions of this questionnaire have been used in several breastfeeding studies (10–13).

Revised text

Maternal recall was assessed using a questionnaire on infant feeding practices. Through an interview, mothers reported infant feeding practices over the previous 24 hours, detailing breast milk intake alongside any liquids, semi-solids, or solids given. Breastfeeding indicators were obtained through interviews with mothers, using a questionnaire adapted from the WHO/UNICEF Indicators for Assessing Infant and Young Child Feeding, Part 2 (24) at the onset of sample collection, just before participants received health services at the facility. The tool is based on current guidelines regarding breastfeeding, complementary feeding, and the feeding of non-breastfed infants and young children under two years of age (25). It is divided into three modules: a household roster, a breastfeeding initiation module (BF), and an infant and young child feeding (IYCF) module. Modified versions of this questionnaire have been used in several breastfeeding studies (16–19) and have adapted the standard WHO "recall since birth" method for local cultural contexts, local practices, and also to confine it within the limits of feeding infants less than the age of six months. EBF was determined by maternal reports confirming no non-breast milk foods/liquids since birth or in the prior 24 hours. This aligned closely with WHO indicators (no liquids/solids beyond breast milk, medicines, or oral rehydration up to the reference age, 6 months), though adaptations addressed local practices.

15. Comment

Lines 156-158: This needs a reference.

Response; We have provided a reference.

Reference cited: McHugh ML. Interrater reliability: the kappa statistic. Biochem Medica. 2012;22(3):276–82.

16. Comment

Line 163: Please clarify what "up to high school education" means. Does this mean that 74.5% reported high school as the highest level of education completed? Does it mean 74.5% reported at least some high school education? Does this mean that 74.5% completed high school or less?

Response

In this case up to high school education means reached and/or completed high school. Some reached but did not complete high school. Others reported that they have reached and completed high school. So, the 74.5% is some or completed high school combined into one group.

Original text

The majority of mothers had up to high school education (74.5%), were unemployed (86.3%) or single (56.9%), as shown in….

Revised text

The majority of mothers reported some or completed high school as the highest level of education (74.5%), were unemployed (86.3%) or single (56.9%), as shown in Table 1.

17. Comment

Figure 1, Model 1: I am surprised to see that EBF was the same in the middle and older age group. I would have expected more of a difference. Have other studies had similar findings?

Response

No, most studies report declining exclusive breastfeeding (EBF) prevalences as infants age from younger (e.g., 0-2 months) to middle/older groups (e.g., 4-6 months) within the 0-6 month period, contrary to the similar EBF prevalences observed in Model 1 (Figure 1). Declining EBF prevalences were, however, found for the less stringent cut-off, which was the more expected trend, and this further supports the conclusion that Model 2 more accurately reflects true EBF than Model 1.

18. Comment

Line 198: On line 156, the authors state that a score of -1.0 is "perfect disagreement" and that 0.21-0.40 [positive] represent fair agreement. So, I am confused that these neg

---

## [Editor Report · Decision Letter 2]

24 Mar 2026

Validating self-reported exclusive breastfeeding in Eswatini using stable isotope techniques

PONE-D-25-44019R2

Dear Dr. Simelane,

We’re pleased to inform you that your manuscript has been judged scientifically suitable for publication and will be formally accepted for publication once it meets all outstanding technical requirements.

Kind regards,

Jennifer Yourkavitch

Academic Editor

PLOS One
---

## [Editor Report · Acceptance letter]

PONE-D-25-44019R2

PLOS One

Dear Dr. Simelane,

I'm pleased to inform you that your manuscript has been deemed suitable for publication in PLOS One. Congratulations! Your manuscript is now being handed over to our production team.

Kind regards,

on behalf of

Dr. Jennifer Yourkavitch

Academic Editor

PLOS One